# Pathogenesis of Alopecia Areata and Vitiligo: Commonalities and Differences

**DOI:** 10.3390/ijms25084409

**Published:** 2024-04-17

**Authors:** Hiroki L. Yamaguchi, Yuji Yamaguchi, Elena Peeva

**Affiliations:** 1Inflammation & Immunology Research Unit, Pfizer, Cambridge, MA 02139, USA; yamaguchi.hi@northeastern.edu; 2Inflammation & Immunology Research Unit, Pfizer, Collegeville, PA 19426, USA

**Keywords:** genome-wide association studies (GWAS), danger-associated molecular pattern (DAMP), MHC class 1 polypeptide-related sequence A (MICA), indoleamine 2,3-dioxygenase (IDO), natural killer cell receptor (NKG2D), interleukin 15 receptor β (IL-15Rβ), hair bulge, hair germ, melanocyte, keratinocyte

## Abstract

Both alopecia areata (AA) and vitiligo are distinct, heterogenous, and complex disease entities, characterized by nonscarring scalp terminal hair loss and skin pigment loss, respectively. In AA, inflammatory cell infiltrates are in the deep reticular dermis close to the hair bulb (swarm of bees), whereas in vitiligo the inflammatory infiltrates are in the epidermis and papillary dermis. Immune privilege collapse has been extensively investigated in AA pathogenesis, including the suppression of immunomodulatory factors (e.g., transforming growth factor-β (TGF-β), programmed death-ligand 1 (PDL1), interleukin-10 (IL-10), α-melanocyte-stimulating hormone (α-MSH), and macrophage migration inhibitory factor (MIF)) and enhanced expression of the major histocompatibility complex (MHC) throughout hair follicles. However, immune privilege collapse in vitiligo remains less explored. Both AA and vitiligo are autoimmune diseases that share commonalities in pathogenesis, including the involvement of plasmacytoid dendritic cells (and interferon-α (IFN- α) signaling pathways) and cytotoxic CD8+ T lymphocytes (and activated IFN-γ signaling pathways). Blood chemokine C-X-C motif ligand 9 (CXCL9) and CXCL10 are elevated in both diseases. Common factors that contribute to AA and vitiligo include oxidative stress, autophagy, type 2 cytokines, and the Wnt/β-catenin pathway (e.g., dickkopf 1 (DKK1)). Here, we summarize the commonalities and differences between AA and vitiligo, focusing on their pathogenesis.

## 1. Introduction

Alopecia areata (AA) and vitiligo are among the skin diseases which highly impact patients’ quality of life. AA and vitiligo are distinct autoimmune diseases characterized by nonscarring hair loss and skin pigment loss, respectively. Genome-wide association studies (GWAS) in AA [1] and vitiligo [2] show quite distinct potential susceptibility/risk genes, indicating that AA and vitiligo are not identical. Additionally, each disease consists of heterogenous populations.

However, AA and vitiligo can coexist, and cumulative evidence shows that both diseases share commonalities in their pathogenesis. AA patients experience a significantly high odds ratio (OR) of developing vitiligo (OR 5.30, 95% CI 1.86–15.10; prevalence 1.1%). Among all of the dermatologic diseases investigated, vitiligo showed the highest OR [3]. Likewise, vitiligo patients experience a significantly high OR of developing AA (OR 2.63, 95% CI 2.50–2.78; prevalence 3.8%). Among all of the dermatologic diseases investigated, AA showed the second highest OR next to psoriasis (OR 3.22, 95% CI 3.07–3.37; prevalence 3.8%) [4]. Atopic dermatitis patients experience a significantly high OR of developing both AA (OR 1.80, 95% CI 1.18–2.76) and vitiligo (OR 2.14, 95% CI 1.29–3.54) [5]. Additionally, as compared to healthy volunteers, a high proportion of patients with AA and vitiligo is positive for autoantibodies against tyrosine hydroxylase, and those autoantibodies from AA and vitiligo patients recognize identical epitopes [6], suggesting the similarity between AA and vitiligo.

Progress has been made in our understanding of the pathogenesis of both diseases and the design of new drugs in recent years. In this review, we summarize the pathogenesis of AA and vitiligo and discuss the commonalities and differences between these two diseases in association with clinical observations.

## 2. Pathogenesis of Alopecia Areata (AA) and Vitiligo

### 2.1. Pathogenesis of Alopecia Areata (AA)

AA is an acquired non-scarring hair loss with or without autoimmune comorbidities including vitiligo, atopic dermatitis, thyroid disease, celiac disease, and type 1 diabetes mellitus [7,8]. AA can be subclassified into patchy AA, alopecia totalis (complete loss of scalp hair), and alopecia universalis (complete loss of hair on the scalp and the body), although clear definitions have not yet been finalized [9]. AA affects up to 2% of the world’s population (lifetime prevalence) depending on the database sources and disease severities [10], without a racial/ethnic [11] or gender/sex preference [12].

Although the complex pathophysiology of AA is not fully understood, cumulative evidence from AA mouse models and human AA skin and blood samples indicates that key players in AA pathogenesis include immune privilege collapse [13], activated interferon-γ (IFN-γ) signaling pathways, and activated pathways involved in the cytotoxic cluster of differentiation (CD)8+ T lymphocytes [14,15,16,17,18,19].

GWAS in AA by Petukhova et al. identified 139 single nucleotide polymorphisms that included cytotoxic T lymphocyte (CTL)-associated antigen 4 (CTLA4), cytomegalovirus UL16-binding protein (ULBP), γ-chain cytokines (e.g., interleukin (IL)-2), and natural killer cell receptor (NKG2D) [1]. The findings suggested that susceptibility/risk biomarkers for human AA include genes associated with autophagy, regulatory T lymphocytes, cytomegalovirus infection, γ-chain cytokines, and, most importantly, autoreactive NKG2D+ CD8+ T lymphocytes. Susceptibility/risk biomarkers also include peroxiredoxin 5 (PRDX5) and syntaxin 17 (STX17) genes associated with hair follicular regulation based on colocalization with keratin 31 that is expressed in the hair shaft and inner root sheath, indicating hair follicular malfunction in AA pathogenesis.

AA anagen hair follicles including the hair bulge (HBg) and the areas adjacent to the dermal papilla (hair bulb (HBb) and supra-bulbar area) express major histocompatibility complex (MHC) class 1 [20] and class 2 with perifollicular infiltrates of T lymphocytes, indicating the collapse of immune privilege [13,21]. Bertolini et al. summarized the recent findings associated with immune privilege and its collapse in AA based on mouse and human studies [22]. Damaged cells release damage-associated molecular patterns (DAMPs) in response to external stimuli including viral infection (e.g., Epstein–Barr virus, hepatitis B/C virus, and SARS-CoV-2) and oxidative stress. DAMPs activate immune cells including antigen-presenting cells/dendritic cells and NK cells and downregulate factors that are associated with the enhancement of immune privilege protection (transforming growth factor (TGF)-β1, TGF-β2, programmed death-ligand 1 (PDL1), IL-10, α-melanocyte-stimulating hormone (αMSH), macrophage migration inhibitory factor (MIF), and indoleamine 2,3-dioxygenase (IDO)) [22,23].

More specifically, MHC class 1 polypeptide-related sequence A (MICA), one of the DAMPs, is a ligand that activates NKG2D+ cells, including cytotoxic CD8+ T lymphocytes, γδ T lymphocytes, and NK cells, thereby contributing to immune privilege collapse, IFN-γ production, and the development of human AA [24]. ULBP3, one of the DAMPs, has a similar role in immune privilege collapse. Impaired autophagy in the HBb area can also be associated with immune privilege collapse [25,26].

Plasmacytoid dendritic cells produce IFN-α, activate cytotoxic T lymphocytes to produce IFN-γ and the chemokine C-X-C motif ligand (CXCL)10, and initiate development of AA lesions in a C3H/HeJ AA mouse model [27]. Plasmacytoid dendritic cells are present in HBb, as measured by myxovirus resistance protein A (MxA; a surrogate marker for local type 1 IFN production), and in the supra-bulbar area, as measured by blood dendritic cell antigen-2 (BDCA-2; CD303; type 2C lectin receptor; plasmacytoid dendritic cell marker) in AA lesions (N = 19) [28], suggesting a role of plasmacytoid dendritic cells in human AA pathogenesis (Figure 1).

Both human and mouse AA skin show increased expression levels of IFN-γ response genes (CXCL9, CXCL10, and CXCL11), CTL-specific transcripts (CD8A, granzyme (GZM)A, and GZMB), and γ-chain cytokines (IL-2 and IL-15) and their receptors (IL-2Rα, IL-15Rα, and IL-15Rβ) [14]. Of note, the expression levels of IL-15 and IL-15Rα (CD215) are colocalized and upregulated in follicular keratinocytes (broadly from HBb extending to close to HBg, in the outer root sheath) of human AA, whereas those of CD8 and IL-15Rβ (CD122) are colocalized and upregulated in autoreactive resident memory T lymphocytes that are adjacent to follicular keratinocytes (close to HBb) [14]. IL-15 activates CD8+ T lymphocytes and NK cells that produce IFN-γ and induces/maintains autoreactive CD8+ T lymphocytes in several diseases, such as celiac disease [29] and multiple sclerosis [30], which is recapitulated in the mouse AA model [14]. Trichohyalin and keratin 16 from keratinocytes can be autoantigens for CD8+ T lymphocytes in AA [31]. However, melanocytes and associated products can also be the target for autoreactive CD8+ T lymphocytes in AA, possibly due to the following two major clinical observations [32,33]. One is that AA does not occur in scalps with gray/white/depigmented hair. The other is that hair regrowth/recovery commonly starts with white/depigmented hair (transient poliosis with thin/vellus hair), followed by the pigmentation from the bottom portion of hair shaft (pigmented terminal hair).

Lee et al. reported that only the depletion of CD8+ T lymphocytes, consisting of five clusters (naïve-like, IFN-γ^high^, CTL-like, resident memory T-like, and chemokine X-C motif ligand 1 (Xcl1)^high^ tumor necrosis factor receptor superfamily 9 (TNFRSF9)^high^), is sufficient to prevent and reverse AA in the graft-induced C3H/HeJ mouse model [34]. They also showed that human AA skin (vs. control skin) is enriched with overall T lymphocytes (23.74% vs. 11.49%), CD8+ T lymphocytes (9.85% vs. 4.11%), regulatory T lymphocytes (2.02% vs. 0.64%), NK T lymphocytes (4.10% vs. 0.85%), and γδ T lymphocytes (2.49% vs. 1.00%), whereas no significant difference is observed in the frequency of CD4+ T lymphocytes between the AA and control skin. Additionally, innate lymphoid cell type 1 (ILC1)-like cells expressing eomesodermin (eomes, required for both ILC1 and NK cell development), CD49a, and NKG2D, may be involved in AA induction via IFN-γ secretion in human [35]. Abnormal interactions between perifollicular mast cells and CD8+ T lymphocytes are also implicated in both mouse and human AA [36].

Recent findings from human skin [37] and blood [37,38,39] also indicate that type 2 cytokines, including IL-13, chemokine C-C motif ligand (CCL)13, CCL17, and CCL18 and dominant signatures of type 2 helper T (Th2)/type 2 cytotoxic T (Tc2) lymphocytes, are involved in AA pathogenesis, explaining the correlation between AA and atopy. AA patients treated with dupilumab, an IL-4Rα monoclonal antibody, demonstrated significant suppression of cellular infiltrates and upregulation of hair keratins in scalp skin biopsies from atopic AA patients [40]. Additionally, dupilumab response was better in AA patients with baseline IgE ≥ 200 IU/mL [41]. These findings indicate the involvement of type 2 cytokines in AA pathogenesis.

Dickkopf 1 (DKK1) is a Wnt inhibitor that suppresses hair morphogenesis in mice [42] and regulates hair follicle (HF) spacing by interacting with Wnt in a reaction–diffusion mechanism [43]. In mice, treatment with DKK1 blocks HF regeneration, whereas treatment with Wnt7a enhances the density of HF regeneration [44]. DKK1 knockout in adipose-derived stem cells results in activation of the Wnt signaling pathway and reduced inflammatory cytokines via the nuclear factor κ-light-chain-enhancer of activated B cells (NF-κB) pathways, thereby promoting hair growth in an AA mouse model [45]. DKK1 is highly expressed in AA scalp skin (N = 31; N = 20) compared to healthy controls (N = 33 [46]; N = 20 [47], respectively), suggesting a role of DKK1 and other Wnt signaling pathways in AA pathogenesis.

### 2.2. Pathogenesis of Vitiligo

Vitiligo is an acquired depigmentation disorder with or without autoimmune comorbidities, such as AA, thyroid disease, psoriasis, inflammatory bowel disease, type 1 diabetes mellitus, and pernicious anemia [48,49,50,51]. Vitiligo can be subclassified into acrofacial, generalized, universal, mixed (accompanied with segmental vitiligo), and rare variants, affecting 0.5% to 2% of the world’s population (lifetime prevalence) depending on the database sources [52], without any racial/ethnic or gender/sex preference [51]. 

The underlying trigger of vitiligo pathogenesis can be oxidative stress causing melanocyte damage, genetics affecting melanocyte growth and differentiation, and autoimmune processes involving autoreactive cytotoxic T lymphocytes, although the complex pathophysiology of vitiligo is not fully understood. Of note, cumulative evidence from vitiligo mouse models [53,54] and human vitiligo skin biopsies [55,56] shows that key players in vitiligo pathogenesis include hyperactive IFN-γ signaling pathways and activated pathways involving cytotoxic CD8+ T lymphocytes [57,58,59,60] (Figure 2).

GWAS from vitiligo patients (mainly from a European-derived white population and Han Chinese population) identified 50 confirmed loci that included CTLA4, interferon regulatory factor 4 (IRF4), MHC class 1 (HLA-A, HLA-B, and HLA-C), MHC class 2 (HLA-DRB1, HLA-DQA1, HLA-DQB1, and HLA-DRA), zinc finger protein Eos (IKZF4; transcription factor expressed in regulatory T lymphocytes), and GZMB [2,61]. These genes associated with immune modulations may serve as susceptibility/risk biomarkers for vitiligo. GWAS also showed that susceptibility/risk biomarkers include genes associated with melanocyte function, indicating its malfunction in vitiligo pathogenesis.

DAMPs produced by damaged melanocytes (as well as IFN-γ produced by autoreactive CD8+ resident memory T lymphocytes) activate antigen-presenting cells/dendritic cells including Langerhans cells and keratinocytes [57,62].

More specifically, under oxidative or heat stress [63,64,65] and possibly infection [63], damaged melanocytes produce heat-shock protein 70 (HSP70), one of the DAMPs. Plasmacytoid dendritic cells produce type 1 IFNs including IFN-α, which induces keratinocytes to secrete CXCL9 and CXCL10, in response to HSP70 [66]. HSP70 also directly activates antigen-presenting cells/dendritic cells and converts T lymphocyte tolerance to autoimmunity (type 1 diabetes mellitus) in mice [67], which is recapitulated in vitiligo. Autophagy is activated in melanocytes and fibroblasts that reside in adjacent vitiligo non-lesions to antagonize degenerative stress, indicating the premature senescent status of these cells [68]. Additionally, impaired autophagy increases CXCL16 secretion from keratinocytes [69], suggesting the involvement of impaired and altered autophagy in vitiligo pathogenesis.

Harris et al. reported that IFN-γ is required for the accumulation of pre-melanosome protein (PMEL)-specific autoreactive CD8+ T lymphocytes in the depigmented lesions of a vitiligo mouse model [54]. PMEL (also known as PMEL17 or glycoprotein 100 (gp100)) is a melanosomal protein that is necessary to maintain the melanosome structure at early stages and which is located within melanocyte cytoplasm among other melanosomal proteins [70]. The genetic background of this vitiligo mouse model results in the presence of interfollicular epidermal (IFE) melanocytes due to overexpression of the tyrosine-protein kinase Kit ligand (cKIT; CD117; stem cell growth factor receptor) within basal keratinocytes [71], whereas normal littermates lack IFE melanocytes and pigmentation. Of note, this mouse model exhibits “patchy” depigmented skin and “spares” the pigmented terminal hair in non-lesion and the terminal hair in lesion after adoptive transfer of PMEL-specific autoreactive CD8+ T lymphocytes, which is clearly recapitulated in human vitiligo without the accompanying AA phenotype [54]. This study indicates that PMEL (or other melanocyte specific proteins) derived from “damaged” melanocytes can serve as an antigen, followed by the recognition of antigen-presenting cells and the generation of melanocyte-specific autoreactive resident memory T lymphocytes in human vitiligo. Additionally, CXCL10, an IFN-γ-specific signature, is sufficient to initiate and maintain depigmentation via the activation of autoreactive CXCR3+ CD8+ T lymphocytes in this vitiligo mouse model [72].

IL-15 (as well as IL-2) activates CD49a+CD8+ resident memory T lymphocytes in human vitiligo skin and induces perforin and GZMB in those cells, followed by melanocyte apoptosis [55]. Those IL-15 activated CD8+ resident memory T lymphocytes produce IFN-γ and TNF-α and express high levels of NKG2D [56]. IFN-α (possibly produced by plasmacytoid dendritic cells) induces the expression of NKG2D ligands, i.e., MICA/MICB, on dendritic cells, thereby maintaining autoimmunity in vitiligo skin lesions [56]. Of note, those melanocyte-specific CD8+ resident memory T lymphocytes, as defined by CD69+ and CD103+ (within the epidermis), also co-express CXCR3, with increased production of IFN-γ (and TNF-α) [73]. Richmond et al. reported that those CD8+ resident memory T lymphocytes express IL-15Rβ (CD122) in human vitiligo skin (whereas keratinocytes express IL-15Rα (CD215),) and that IL-15Rβ blockade reverses vitiligo in mouse model [74]. Richmond et al. also reported that autoreactive CD8+ resident memory T lymphocytes require a self-antigen (PMEL) and produce IFN-γ in a mouse model [75]. They also reported that PMEL-specific resident memory T lymphocytes produce CXCL9 and CXCL10 and possess a sensing/alarm function, thereby recruiting circulating memory T lymphocytes that are required to maintain the depigmentation in a mouse model.

IFN-γ has an additional effect on vitiligo pathogenesis. Type 1 cytokines IFN-γ and TNF-α induce melanocyte detachment via E-cadherin disruption and the release of its soluble form, possibly due to increased matrix metalloproteinase 9 (MMP9) in the skin and blood in vitiligo patients [76], contributing to vitiligo pathogenesis. Discoidin domain receptor 1 (DDR1) forms complexes with E-cadherin [77] and acts as a collagen IV adhesion receptor, thereby regulating the location of melanocytes at the basement membrane zone [78]. As genetic variants of the DDR1 gene are observed in Brazilian [79] and Korean [80] populations, dysfunction of DDR1 may be associated with vitiligo pathogenesis, although further functional analysis is necessary [81]. Bastonini et al. also summarizes the involvement of keratinocytes, fibroblasts, and the extracellular matrix in vitiligo pathogenesis [82].

Additionally, blood analyses by Czarnowicki et al. indicated that Th2/Tc2 lymphocyte signatures are dominant in vitiligo patients (N = 19) in a similar extent to AA patients (N = 32) and atopic dermatitis patients (N = 43) [38]. They also reported that Th1/Tc1, Th9/Tc9, Th17, and Th22 lymphocyte signatures are dominant in vitiligo patients, whereas regulatory T lymphocyte signatures are recessive [38]. A systematic review to investigate pathogenesis shows that vitiligo biomarkers include CD4, CD8, CXCL9, and nucleotide-binding oligomerization domain-like receptor family pyrin domain containing 1 (NLRP1) in skin and IL-1β, IL-17, IFN-γ, TGF-β, autoantibodies, oxidative stress biomarkers (e.g., reactive oxygen species), regulatory T lymphocytes, soluble (s)CD25, sCD27, CXCL9, and CXCL10 in blood, although further investigations are required [83]. More recent meta-analysis data show that CXCL9, CXCL10, CCL5, CXCL8 (IL-8), CXCL12, and CXCL16 can serve as diagnostic blood biomarkers for vitiligo [84].

DKK1 is expressed more highly in acral fibroblasts than non-acral fibroblasts and suppresses melanocyte function, explaining why the palms of the hands and soles of the feet (palmoplantar/ volar skin) are hypopigmented in human [85]. DKK1 also suppresses melanin distribution from melanocytes to keratinocytes via the suppression of proteinase-activated receptor 2 (PAR2), a melanin receptor expressed on keratinocytes, based on the comparison between palmoplantar skin and non-palmoplantar skin in human [86]. DKK1 inhibits epidermal melanocyte proliferation after wounding in mice [87]. DKK1 is highly expressed in the dermis of vitiligo lesions [88,89] and cultured fibroblasts from vitiligo lesions [90], suggesting a role of DKK1 and other Wnt signaling pathways in vitiligo pathogenesis [91].

## 3. Discussion

### 3.1. Commonalities in Pathogenesis between Alopecia Areata (AA) and Vitiligo

Findings from GWAS indicate that AA and vitiligo are distinct, heterogenous, and complex disease entities, as described in Section 2.1 and Section 2.2. For example, TNF-α 308 G/A polymorphism (rs1800629) can be a susceptibility/risk biomarker for vitiligo, but not for AA, in Egyptian patients [92]. However, Silverberg reports that both AA and vitiligo share the following susceptibility/risk gene alleles: (1) tyrosine-protein phosphatase nonreceptor 22 (PTPN22), (2) Fas ligand (FASLG), (3) CTLA4, (4) HLA-A, (5) HLA-DRB1/DQA1, (6) IL-2Rα, (7) CD44, (8) IKZF4, and (9) Src-homolog 2B adaptor protein 3 (SH2B3) [93]. This indicates that, at the very least, common pathways leading to AA or vitiligo exist.

Several candidates can serve as diagnostic biomarkers for both AA and vitiligo. Serum CXCL9 (AA > vitiligo) and CXCL10 (AA = vitiligo) are elevated in both AA (N = 15) and vitiligo (N = 15) as compared with controls (N = 15) [94]. Serum IFN-γ is also elevated in AA (N = 33) and vitiligo (N = 30) [95], suggesting the involvement of IFN-γ-driven immune responses in both diseases. Serum IL-1β and serum IL-6 (vitiligo > AA) are also elevated in both diseases [95], supporting the hypothesis that oxidative stress is associated with the promotion and amplification of inflammatory process in both AA and vitiligo. Serum brain-derived neurotrophic factor (BDNF) and serum vitamin D are decreased both in AA (N = 30) and vitiligo (N = 30), as compared with controls (N = 30) [96]. Serum MIF is elevated both in AA (N = 22) and vitiligo (N = 20), as compared with controls (N = 20), and correlates with disease severity [97].

Although inflammatory cell infiltrates (CD4+ and CD8+ T lymphocytes) are observed in both AA and vitiligo lesions, those inflammatory cells are less densely populated than in other inflammatory skin diseases, like atopic dermatitis and psoriasis [98]. As described above in Section 2.1 and Section 2.2. and reviewed by other investigators [98,99,100,101], commonalities in pathogenesis between AA and vitiligo include (1) enhanced type 1 IFN production by plasmacytoid dendritic cells in response to DAMPs, (2) enhanced IFN-γ signaling pathways, including CXCL9 and CXCL10, (3) CD8+ T lymphocytes activation, and (4) involvement of (a) oxidative stress, (b) autophagy, (c) type 2 cytokines, and (d) Wnt/β-catenin signaling pathways, including DKK1. Of note, cumulative evidence from human skin and blood strongly supports the involvement of type 2 cytokines in AA pathogenesis [37,38,39,40,41]. Additional research may be required to support the link between vitiligo and atopy/type 2 cytokines, but blood gene analyses show the dominance of type 2 cytokine signatures in vitiligo patients in a similar extent to AA and atopic dermatitis patients [38]. Additionally, atopic dermatitis patients are at high risk of developing both AA and vitiligo [5].

The alopecia areata disease severity index (ALADIN) is a gene expression metric that consists of the following three components: (1) cytotoxic T cell infiltration markers, (2) IFN-γ-associated markers, and (3) hair keratin markers [15]. Since AA and vitiligo share the first two components, a similar approach may be feasible to assess vitiligo severity by replacing hair keratin markers with melanocyte markers.

### 3.2. Differences in Pathogenesis between Alopecia Areata (AA) and Vitiligo

The differences in pathogenesis between AA and vitiligo include (1) inflammatory cell infiltration location (the deep reticular dermis close to HBb (swarm of bees) in AA vs. the papillary dermis and/or epidermis possibly close to epidermal melanocytes in vitiligo), (2) the identified DAMPs (MICA and ULBP3 in AA vs. HSP70 in vitiligo; the downstream pathways are similar), and (3) robust evidence of immune privilege collapse in AA (Figure 3). 

Immune privilege is defined as a protective mechanism of stem cells against autoimmunity and inflammation that is maintained via local production of immunoregulatory cytokines (as listed in Section 2.1) and low expression levels of MHC class 1. Immune privileged sites include not only HF but also bone marrow, serving as a niche for hematopoietic stem cells via forkhead box P3 (FOXP3)+ regulatory T lymphocytes [102]. Vitiligo-like lesions occur with an incidence of up to 25% under anti-PD1 (e.g., pembrolizumab [103] and nivolumab [104]), PDL1 (e.g., atezolizumab), and CTLA4 (e.g., ipilimumab) treatments that target immune privilege checkpoints for advanced melanoma and other cancers [105,106]. Although there is no direct evidence of immune privilege collapse in vitiligo, it may be hypothesized that immune privilege collapse occurs in only a small portion of the lower HBg and hair germ, where melanocyte stem cell precursors reside, leading to (1) enhancement of IFN-γ signaling pathways and (2) exposure of self-antigens to cytotoxic T lymphocytes, thereby causing subsequent melanocyte instability in vitiligo pathogenesis, as Boniface et al. reported [107]. Further studies are needed to confirm or refute the role of MIF in vitiligo pathogenesis via immune privilege collapse in the same way as in AA pathogenesis [97].

## Figures and Tables

**Figure 1 ijms-25-04409-f001:**
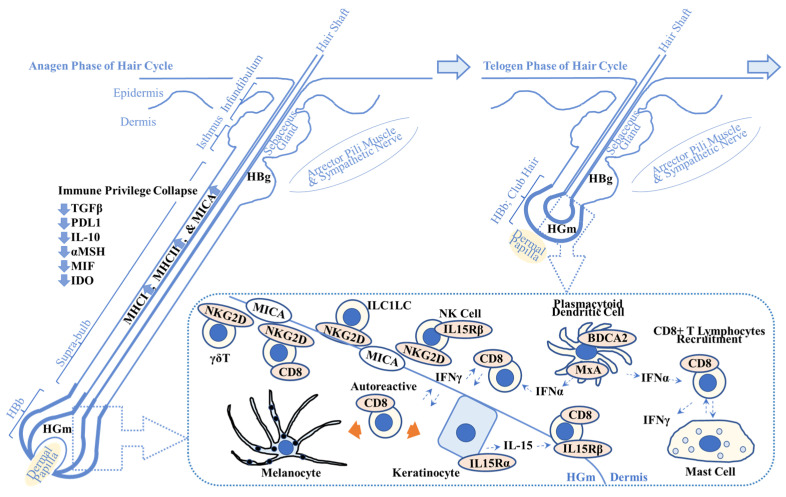
Overview of possible alopecia areata pathogenesis. This scheme shows the key players in alopecia areata (AA) pathogenesis. Immune privilege collapse occurs due to downregulation of immunosuppressive factors including transforming growth factor β (TGFβ), programmed death-ligand 1 (PDL1), interleukin-10 (IL-10), α-melanocyte stimulating hormone (αMSH), migration inhibitory factor (MIF), and indoleamine 2,3-dioxygenase (IDO). It results in the enhanced expression of major histocompatibility complex class 1 (MHCI) and class 2 (MHCII) and MHCI polypeptide-related sequence A (MICA), one of the damage-associated molecular patterns. (**Dotted box**) MICA recruits natural killer cell receptor G2D (NKG2D)+ cells including cluster of differentiation 8 (CD8)+ cytotoxic T lymphocytes, NK cells, γδ T lymphocytes, and innate lymphoid cell type 1-like cells (ILC1LC). Plasmacytoid dendritic cells produce interferon-α (IFN-α) that activates CD8+ T lymphocytes. IFN-γ produced by CD8+ T lymphocytes recruits CD8+ T lymphocytes to the AA lesion. Keratinocytes produce IL-15 that recruits CD8+ T lymphocytes via IL-15 receptor β (IL15Rβ). Autoreactive T lymphocytes target melanocytes and their associated products (**left orange arrow**) and keratinocyte-derived products (**right orange arrow**). Abnormal interactions between CD8+ T lymphocytes and mast cells are implicated in AA pathogenesis. Refer to Section 2.1. for details. **HBg** = hair bulge (stem cell reservoir for keratinocytes (middle and top) and melanocytes (bottom)); **HGm** = hair germ (stem cell reservoir for melanocytes); HBb = hair bulb; MxA = myxovirus resistance protein A; BDCA2 = blood dendritic cell antigen-2.

**Figure 2 ijms-25-04409-f002:**
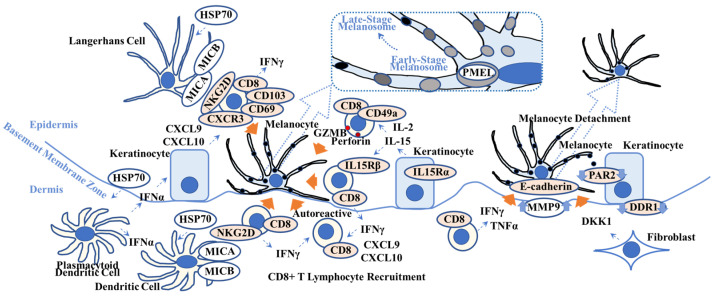
Overview of possible vitiligo pathogenesis. This scheme shows key players in vitiligo pathogenesis. Heat-shock protein 70 (HSP70), one of the damage-associated molecular patterns, activates Langerhans cells and plasmacytoid dendritic cells that produce interferon-α (IFNα). IFNα stimulates keratinocytes to produce chemokine C-X-C motif ligand 10 (CXCL10) that activates CXCR3+ CD8+ T lymphocytes. IL-15 activates CD49a+ CD8+ T lymphocytes and induces granzyme B (GRMB) and perforin. The MHC class 1 polypeptide-related sequence A/B (MICA/MICB) on dendritic cells maintains natural killer cell receptor G2D (NKG2D)+ CD8+ T lymphocytes in the vitiligo lesion. IFNγ not only recruits CD8+ autoreactive T lymphocytes but also induces melanocyte detachment via E-cadherin disruption caused by an increase in matrix metalloproteinase 9 (MMP9). Discoidin domain receptor 1 (DDR1) forms a complex with E-cadherin, indicating its involvement in vitiligo pathogenesis. Dickkopf 1 (DKK1) decreases melanocyte function and proliferation and melanin uptake by keratinocytes via suppressed proteinase-activated receptor 2 (PAR2). Orange arrows indicate melanocyte apoptosis (**all 5 left arrows**), melanocyte detachment (**second right arrow**), and suppression of melanocyte function (**far right arrow**). (**Dotted box**) Pre-melanosome protein (PMEL) can be a target for autoreactive CD8+ T lymphocytes. Refer to Section 2.2. for details. TNFα = tumor necrosis factor-α; IL15R = interleukin-15 receptor; CD = cluster of differentiation.

**Figure 3 ijms-25-04409-f003:**
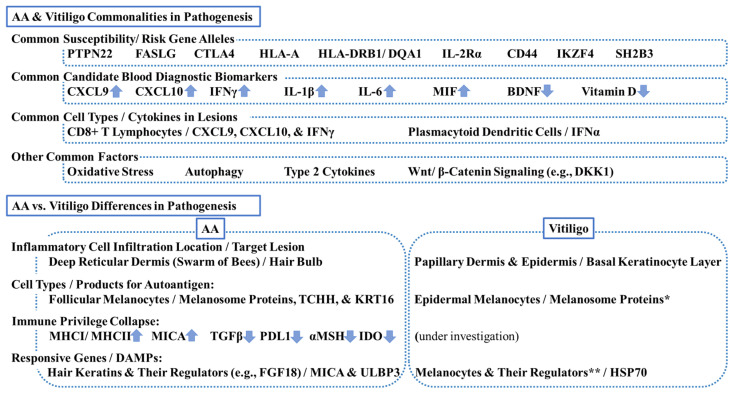
Commonalities and differences in pathogenesis between alopecia areata (AA) and vitiligo. This scheme summarizes key commonalities and differences. PTPN22 = tyrosine-protein phosphatase nonreceptor 22; FASLG = Fas ligand; CTLA4 = cytotoxic T lymphocyte-associated antigen 4, HLA = human leukocyte antigen; IL-2R = interleukin-2 receptor, CD = cluster of differentiation; IKZF4 = zinc finger protein Eos; SH2B3 = Src-homolog 2B adaptor protein 3 CXCL = chemokine C-X-C motif ligand; IFN = interferon; MIF = macrophage migration inhibitory factor; BDNF = brain-derived neurotrophic factor; DKK1 = dickkopf 1; TCHH = trichohyalin; KRT = keratin; MHC = major histocompatibility complex; TGF = transforming growth factor; PDL1 = programmed death-ligand 1; MSH = melanocyte stimulating hormone; IDO = indoleamine 2,3-dioxygenase; FGF18 = fibroblast growth factor 18; MICA = MHCI polypeptide-related sequence A; ULBP3 = cytomegalovirus UL16-binding protein; HSP70 = heat-shock protein 70. * Melanosomal proteins can be autoantigens for both AA and vitiligo. ** Melanocyte regulators can be responsive genes for both diseases.

## Data Availability

Not applicable.

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
