# Peer review of "Pathogenesis of Alopecia Areata and Vitiligo: Commonalities and Differences"

_ijms, 2024, doi:10.3390/ijms25084409_

Round 1

Reviewer 1 Report

Comments and Suggestions for Authors

Dear Authors,

I read your paper with interest. Writing about two well know conditions with similar autoimmune explanations.

The theory of an immune privilege collapse for AA is well known. You wanted to see whether this could also be the case for Vitiligo. But with my own experience with this disease and its treatment it could be considered unlikely that a Vitiligous area was before an immune privileged area. Since allograft rejection occurs not only in the vitiligo but also in the not vitiliginous skin.

There are however stemcells surviving in an immune protected area in the hair bulbus within the vitiligo and these may give rise to new melanocytes repopulating the vitiligenous skin That could also mean that the surrounding skin was not IP. (there was thus in a Vitiligenous area an small IP area)

Some more specific comments:

Fig 1: This is an could be an explanation of AA.                                                  AA arises because certain receptors become available and certain mediators for these receptors are produced . Other receptors and mediators become available too, because the factors that give rise to AA are released and may stimulate their visibility.  Both groups are demonstrable, the authors indicate them but do not separate them.                                                                           It is also important what are the triggers, infections stress other…                Indeed, it is just an overview of what is described in AA.                                 What they describe in the text is only what could be possible.

Fig 2: Again, a nice picture but again a “could be”.                                              In my experience indeed HSP 70 could indeed be seen as a possible trigger induced by infections.

The discussion:                                                                                                     Fig 3 gives the observations clearly.                                                                       I  agree there is no direct evidence for IP collapse in vitiligo.  But not with the speculation thereafter .

Vitiligo like lesions occur with an incidence of up to 25% under anti-PD1 (including pembrolizumab), PDL1, and CTLA4 treatments that are targeting immune privilege checkpoints for advanced melanoma. Probably this initiated the interest of the authors.

Author Response

Dear Reviewer 1,

Please find our responses to your comments as attached. Thank you for your time and consideration.

Sincerely,

Yuji Yamaguchi, MD, PhD

Reviewer 2 Report

Comments and Suggestions for Authors

AA and Vitiligo are among the diseases that leave a psychological mark on patients. In recent years, great progress has been made in the field of research on the pathogenesis of both diseases and the design of new drugs. There is the reason why, I consider the topic of the work to be important and up-to-date.

Specific comments as requested
What is the main question addressed by the research?
This is an interesting manuscript in which authors try to explain the reasons for the frequent co-occurrence of two different diseases, which are Alopecia Areata and Vitiligo. In this review not only the pathogenesis of the both diseases but also clinical similarities and differences are discussed.

2. What parts do you consider original or relevant for the field? What specific gap in the field does the paper address?
The most interesting parts of the manuscript are connected with the immunological abnormalities in above diseases. The results of the latest studies revealed that human AA skin is enriched with overall T lymphocytes , CD8+, regulatory and NK T lymphocytes. Moreover, recent findings from human skin and blood also indicated the involvement of type 2 cytokines in AA pathogenesis what explains  the correlation between AA and atopy.

3. What does it add to the subject area compared with other published material?
The review paper is based on the recently published literature.

4. What specific improvements should the authors consider regarding the
methodology? What further controls should be considered?
No suggestions

5. Please describe how the conclusions are or are not consistent with the
evidence and arguments presented. Please also indicate if all main questions
posed were addressed and by which specific experiments.

The conclusions are consistent with the evidence and arguments presented and all main questions posed were addressed.

6. Are the references appropriate?
The work is well written on the basis of the latest literature, apart from a few editorial errors, it is correct in terms of language and style. In my opinion, it can be published in the JCM.  

7. Please include any additional comments on the tables and figures and quality of the data.
Figures 1, 2 and 3 graphically represent an overview of AA and Vitiligo pathogenesis, and show the differences between those two diseases. They are comprehensible, clear and well-structured

Author Response

Dear Reviewer 2,

Please find our responses to your comments as attached. Thank you for your time and consideration.

Sincerely,

Yuji Yamaguchi, MD, PhD
